# OpenReview forum: "Effects of Scale on Language Model Robustness"
_ICLR.cc/2025/Conference — Submitted to ICLR 2025_

### Official Review · Reviewer_qvqy · 2024-10-31

**Soundness:** 3
**Presentation:** 2
**Contribution:** 2
**Rating:** 5
**Confidence:** 4

**Summary:**

This paper studies the robustness of language models in the classification setting, comparing the impact of adversarial training with model scaling as defense strategies.
The authors argue that adversarial training may offer a more compute-efficient solution compared to scaling up the model size.

**Strengths:**

- Originality: To my knowledge, this paper presents novel findings. I am not aware of any other work exploring the trade off in increasing models size and adversarial training for language classification systems.

- Quality: The writing is clear and the findings are presented nicely.

- Experimental Design: The empirical findings are clear and make a succinct point about compute tradeoffs and the experiments are principled.

**Weaknesses:**

1. **Narrow Experiments** Only Pythia models are considered. Since those models are designed for studying model scaling, they have lots of hyperparameters held constant across model sizes. We know these are not the optimal hyperparameters in most cases, it would be much stronger if the paper included results on newer and stronger models. Since the claim is that robustness doesn't reliably improve with scale, one might wonders how the results look on the llama family or the Qwen models.

2. **Unoriginal Claims** The strong final claim in the abstract is that adversarial training alone isn't enough to solve robustness, but this is well accepted by the adversarial robustness community. It feels less like a novel claim to make and more like a commonly held belief to state, especially with only narrow empirical results to back it up.

3. **limited impact** The impact of adversarial attacks on text classifier may be hard to measure directly, but seems limited to me. In practice a spam filter, for example, might have many components in addition to an LLM and so I'm unconvinced that the results in this paper imply any practical risk. Now one might make a similar argument across the robustness space, but often there is more to cling to. For example, with image classifiers, we have no better means than deep learning and thus its vulnerabilities are critical to image processing systems. Another example with generative language models is jailbreaking -- perhaps also of limited impact -- the goal there is to show that alignment is weak or penetrable and alignment is an entire portion of model building wherein harmful behavior specifically is meant to be reduced.

4. **Odd baseline** To my best understanding, the goal of this paper is to study mitigation of adversarial vulnerability of text classifiers, but the only two defenses considered are scaling up and adversarial training. Scaling, to my knowledge, is not often used as a baseline for defenses. And although I can appreciate that if robustness was an emergent (arrives with increased scale) property that would be important to document, the finding in this paper is that scaling up isn't a great defense.

Summary of weaknesses: In all, I think the threat model is less than compelling in light of all the existing work on classifier robustness and the limited scope of experiments doesn't make up for that. Furthermore, the assumption/intuition that scale is an adversarial defense seems under-motivated to me. My questions below extend these weaknesses.

___

**Minor points not affecting score:**
1. Lines 51 and 52 in the pdf looks like a typo -- not sure what that sentence is meant to say.
2. The details of how much data is used for finetuning, how long computing attacks strings takes, when is the GCG optimization stopped (at the first point that it fools the classifier?) should all be in the body of the paper. I'm not concerned that this isn't reproducible per se, but I have played with GCG and model finetuning and I find those details values not obvious and their absence from the main text of the paper curious.

**Questions:**

1. **Narrow Experiments** How to the results look for other models like the llama family or the Qwen models?

2. **Unoriginal Claims**  Can the authors expand on the claims? Is there an intuition that might help someone familiar with adversarial robustness understand the starting place that there might be a chance adversarial training would work here? In other words, why isn't the failure of adversarial training to "sufficient[ly] solve robustness" the a priori expectation?

3. **limited impact** Can the authors offer more motivation to the work? For example, are there spam filters in practice that comprise only a finetuned LLM for classification? This seems rather limited to me but I am not an expert on classification based applications.

4. **Odd baseline** How would baseline defenses like those proposed by Jain et al. (2023) work here? The perplexity filter in particular is dismissed in the related work section because stronger attacks exist, but the attacks used in this paper's experiments are not those specific stronger attacks. The GCG strings are likely to be caught by any perplexity filter or bot detection method. Thus, how can we conclude that these text classifications systems are vulnerable?

---

> ### Author Response · Authors · 2024-11-16
>
> Thank you for your helpful comments. We have written out a [response to all reviewers](https://openreview.net/forum?id=IAFLoDz6H5&noteId=GK46muABVY) and [plan for the rebuttal period](https://openreview.net/forum?id=IAFLoDz6H5&noteId=w0Lg16u4fn), and would welcome any feedback. We respond to your specific questions and comments below.
>
> > Only Pythia models are considered. Since those models are designed for studying model scaling, they have lots of hyperparameters held constant across model sizes. We know these are not the optimal hyperparameters in most cases, it would be much stronger if the paper included results on newer and stronger models. Since the claim is that robustness doesn't reliably improve with scale, one might wonders how the results look on the llama family or the Qwen models.
>
> > How to the results look for other models like the llama family or the Qwen models?
>
> As you mention, we chose Pythia as the most modern model family where we could study scaling over 3 orders of magnitude. We agree that it would be valuable to check that our results hold for other model families, so we are training Qwen on our tasks and expect to have results ready during the rebuttal period.
>
> > The strong final claim in the abstract is that adversarial training alone isn't enough to solve robustness, but this is well accepted by the adversarial robustness community. It feels less like a novel claim to make and more like a commonly held belief to state, especially with only narrow empirical results to back it up.
>
> > Can the authors expand on the claims? Is there an intuition that might help someone familiar with adversarial robustness understand the starting place that there might be a chance adversarial training would work here? In other words, why isn't the failure of adversarial training to "sufficient[ly] solve robustness" the a priori expectation?
>
> Thanks to the reviews, we have realized that our original framing obscured the interaction between scale and adversarial robustness. As seen in Figure 6, scaling up the compute used in adversarial defenses can in fact greatly decrease the attack success rate. It is only when accounting for the attacker’s ability to scale up compute that adversarial training is shown to be insufficient—if the defender doubles the adversarial training compute, then the attacker needs to less than double their adversarial attack compute to maintain the same attack success rate.
>
> Our experiments and analysis provide unique perspectives on adversarial training. Given enough compute, adversarial training does work, but it’s not effective enough relative to an attacker who can also scale compute.
>
> > The impact of adversarial attacks on text classifier may be hard to measure directly, but seems limited to me. In practice a spam filter, for example, might have many components in addition to an LLM and so I'm unconvinced that the results in this paper imply any practical risk. Now one might make a similar argument across the robustness space, but often there is more to cling to. For example, with image classifiers, we have no better means than deep learning and thus its vulnerabilities are critical to image processing systems. Another example with generative language models is jailbreaking -- perhaps also of limited impact -- the goal there is to show that alignment is weak or penetrable and alignment is an entire portion of model building wherein harmful behavior specifically is meant to be reduced.
>
> > Can the authors offer more motivation to the work? For example, are there spam filters in practice that comprise only a finetuned LLM for classification? This seems rather limited to me but I am not an expert on classification based applications.
>
> Thank you for this comment; it has become clear to us that we did not sufficiently justify the approach taken in the paper. Please see our [response to all reviewers](https://openreview.net/forum?id=IAFLoDz6H5&noteId=GK46muABVY) for a more complete explanation of what we were trying to do.
>
> To respond directly here too: the main focus of our paper is not about whether or not systems are vulnerable—indeed, we know that even frontier chatbot models are vulnerable. Instead, we are trying to understand the impact of scale on adversarial robustness in general. We choose the classification for several reasons; one important reason was that it enabled us to run experiments over 3 orders of magnitude of model size in a way that was not possible with generation.
>
> Note that we are additionally working on a generative experiment using Qwen to ensure that the behavior observed in classification applies to generation as well.
>
> [continued in next comment]

---

> ### Author Response · Authors · 2024-11-16
>
> [continued]
>
> > To my best understanding, the goal of this paper is to study mitigation of adversarial vulnerability of text classifiers, but the only two defenses considered are scaling up and adversarial training. Scaling, to my knowledge, is not often used as a baseline for defenses. And although I can appreciate that if robustness was an emergent (arrives with increased scale) property that would be important to document, the finding in this paper is that scaling up isn't a great defense.
>
> > How would baseline defenses like those proposed by Jain et al. (2023) work here? The perplexity filter in particular is dismissed in the related work section because stronger attacks exist, but the attacks used in this paper's experiments are not those specific stronger attacks. The GCG strings are likely to be caught by any perplexity filter or bot detection method. Thus, how can we conclude that these text classifications systems are vulnerable?
>
> We assume you are referring to https://arxiv.org/pdf/2309.00614, and that the baseline methods are thus perplexity filtering, paraphrasing, retokenization, and adversarial training? We explored the use of a perplexity filter early on in the project, finding it was effective against adversarial suffixes of length 10 (which appears to be a standard and thus is the length we use in this paper). However, decreasing adversarial suffix length to 3 still yielded a successful attack which we were unable to catch even when decreasing the perplexity filter window. Given the similarity of results between RandomToken and GCG, we expect that other attacks such as AutoDAN (which circumvents perplexity filtering) would exhibit similar behavior (https://arxiv.org/abs/2310.04451), though we have not tested it.
>
> We agree that paraphrasing would be an interesting defense to study in the context of scaling. As it stands, including it would add significant complexity to the project we decided to postpone it for further work. We feel similarly about retokenization: it is an interesting defense, but beyond the scope of the current paper.
>
> As a general comment, we would mention that for the past decade the robustness community has been trying to make computer vision models robust, without success. Similarly, for the past few years we have been trying to make language models robust, without success. Since we wanted to focus on the effects of scale on robustness—independent of the details of any specific attack or defense—we decided to use simple and powerful techniques (GCG, adversarial training) in order to draw general conclusions. Please also see our response sent to all reviewers for more details on our framing and intentions for more background on this.
>
> > In all, I think the threat model is less than compelling in light of all the existing work on classifier robustness and the limited scope of experiments doesn't make up for that. Furthermore, the assumption/intuition that scale is an adversarial defense seems under-motivated to me. [...]
>
> Thank you for this comment—it makes clear to us that we were not explicit enough in the framing of our experiments. Going into the project, we were agnostic regarding the extent to which model size would confer adversarial robustness, however other authors have certainly suggested that it does (e.g., Figure 17 of https://arxiv.org/pdf/2210.14891, Figure 7 of https://openreview.net/pdf?id=HyxJhCEFDS). We cite several other opinions on this in our introduction.
>
> > Lines 51 and 52 in the pdf looks like a typo -- not sure what that sentence is meant to say.
>
> You are of course correct! We have fixed this, thank you for pointing it out.
>
> > The details of how much data is used for finetuning, how long computing attacks strings takes, when is the GCG optimization stopped (at the first point that it fools the classifier?) should all be in the body of the paper. I'm not concerned that this isn't reproducible per se, but I have played with GCG and model finetuning and I find those details values not obvious and their absence from the main text of the paper curious.
>
> Thank you for this point. We were bumping up against the page limit but will relegate the RandomToken to GCG transfer experiment to the appendix to make room for these details in the main body. We will also open-source the codebase in the coming months which should help a lot with reproducibility!
> To briefly address your specific questions here as well:
> * For each task, we finetune on 20,000 datapoints for 3 epochs.
> * In adversarial training, computing a successful attack string takes 98% of our compute time (the remaining 2% is training on the attacked datapoints). In absolute terms, for the largest models, it takes up to 10 hours on 2 H100 GPUs to compute the attack strings for 1 adversarial training round.
> * For each datapoint, we run GCG for the full number of iterations, and use the attack string found in the final iteration.

---

> ### Author Response · Authors · 2024-12-02
> **Following up on new results**
>
> We have updated our paper with new experimental results—see [here](https://openreview.net/forum?id=IAFLoDz6H5&noteId=9sGAdINuRf) for a full description.
>
> In the new version of the paper, we more clearly motivate our design decisions, and perform several additional experiments.
> Specifically, we reproduce our experiments on the [Qwen2.5](https://github.com/QwenLM/Qwen2.5) Base model family, we adversarially attack Qwen2.5 Instruct on a generative task ([StrongREJECT](https://arxiv.org/pdf/2402.10260])), and we implement a new state-of-the-art black box attack ([BEAST](https://arxiv.org/pdf/2402.15570)).
>
> We hope that the clearer framing and our new results on Qwen address your concerns.
>
> Please let us know if you have additional comments or suggestions. Thank you.

---

> > ### Comment · Reviewer_qvqy · 2024-12-02
> > **Thoughts on the new results**
> >
> > I listed 4 weaknesses and asked 4 questions and the authors have addressed some of these so I will raise my score from 3 to 5. I am still not convinced this paper should be accepted, here is an updated review:
> >
> > 1. Narrow Experiments --- this was addressed by including Qwen models and new tasks. (More is more here, and if the authors continue their work in this direction I'd advocate for more models/tasks etc.)
> > 2. Unoriginal Claims --- the authors have improved the framing and I see a push now for a direction other than "adversarial training isn't enough" but I'm not satisfied that the new claims are particularly strong or novel. For example, the trends with model size are very weak in the new (very pretty and clear) plots. The idea that scale alone isn't enough for robustness seems well documented here but not really novel. Furthermore the claim the "offense is winning" seems very consistent with all the work in the field and the general consensus among the community that we haven't solved adversarial robustness.  More examples with quotes:
> >     a. "Offense widens its advantage" as scale increases does seem novel to me. As in, I don't know any other work that makes this claim.
> >     b. "Larger models benefit more from safety-training" seems less novel given the results across adversarial LLM papers where SOTA large models tend to be more robust than small/weak models.
> > 3. Limited impact --- partly addressed by including a generative task (some information on this task is still missing, for example Appendix A has no samples or detail on the StrongREJECT data. I know this is available in the cited work, so this remains a minor point, but worth noting).
> > 4. Odd baseline --- I don't think this was addressed well. Any work on adversarial defenses needs to consider the baselines. I think the new framing of this paper omits the advice part, which is has to since it lacks the comparisons. So what would the authors recommend to a model trainer before they allocate compute/energy to model scale and adversarial training? How much of the defense could be gained with a compute-cheap method like a perplexity filter or a paraphrasing defense or even smoothing (like SmoothLLM https://arxiv.org/abs/2310.03684)? Without these baselines the concluding points and several claims in the paper are missing some support.
> >
> > I do appreciate all the work, the authors have clearly worked very hard during this rebuttal and the work has improved, so I am increasing my score. The weaknesses listed in this updated review still keep me from recommending that we accept the paper in its current form.

---

> ### Author Response · Authors · 2024-12-03
>
> > 3. Limited impact --- partly addressed by including a generative task (some information on this task is still missing, for example Appendix A has no samples or detail on the StrongREJECT data. I know this is available in the cited work, so this remains a minor point, but worth noting).
>
> Thank you for bringing this to our attention, we will be sure to add a description and some example datapoints to the Appendix in the next version.
>
> > 4. Odd baseline --- I don't think this was addressed well. Any work on adversarial defenses needs to consider the baselines. I think the new framing of this paper omits the advice part, which is has to since it lacks the comparisons. So what would the authors recommend to a model trainer before they allocate compute/energy to model scale and adversarial training? How much of the defense could be gained with a compute-cheap method like a perplexity filter or a paraphrasing defense or even smoothing (like SmoothLLM https://arxiv.org/abs/2310.03684)? Without these baselines the concluding points and several claims in the paper are missing some support.
>
> Thank you for elaborating on this—your point is well-taken. It’s true that we do not consider baseline defenses in the paper and that this weakens the message of the paper as written.
>
> We comment on the baselines you mention individually below:
>
> ### Perplexity filtering
> While we do not show perplexity results in the paper itself, this was in fact something that we explored earlier in the project. We found that, for the GCG attack, perplexity filtering was able to flag close to 100% of attacked examples while only filtering out 1% of clean examples when we used an adversarial suffix of length 10 (which is what we use for RandomToken and GCG in the paper). However, when we used an adversarial suffix of length 3—which we found to be almost as effective for the attacker—we were unable to find a sliding window size and perplexity threshold that could filter out the attacked examples while not also filtering out a large proportion of the clean examples.
>
> To address your perplexity concerns for our attack implementation, we have now checked the efficacy of a sliding window based perplexity filter on all successful attacks within our logs of 60 examples across the first 3 adversarial training rounds using the BEAST attack on the Spam and Harmless datasets, with 7 successful attacks on the Spam classifier and 14 successful attacks on the Harmless classifier.
>
> We find that for sliding window sizes of 3, 7, 10, and 20 tokens, no Spam attack strings could be identified with a perplexity filter, and at most 2/14 of the Harmless attack strings could be identified with a perplexity filter which was set to the optimal per-example perplexity threshold. Specifically, for every Spam task example and in 12/14 of the Harmless task examples, the maximum n-token sliding window perplexity with any portion of the adversarial suffix was lower than the maximum n-token sliding window perplexity across the non-attacked portion of the string, such that setting any n-token perplexity threshold low enough to catch the attack string would have yielded a false positive on the non-attacked portion of the string.
>
> Graphs (and code) here: [https://anonymous.4open.science/r/anon-beast-perplexity-D37C/](https://anonymous.4open.science/r/anon-beast-perplexity-D37C/)
>
> ### Paraphrasing
>
> We consider paraphrasing to be a relatively ineffectual defense which is unlikely to be used in practice for two reasons. Firstly, it adds a substantial latency cost in order to rephrase the prompt. Secondly, it likely leads to negative performance impacts on more complex tasks where people are likely to want to use more advanced models.
>
> ### Smoothing
>
> We are also not confident that smoothing is a realistic defense for one to use. Although SmoothLLM does not require any additional train-time compute, it does require running each query in parallel at test-time. A multiplicative factor on test-time compute is a hard sell for the company deploying a model—we believe that economic incentives will push AI API providers away from this kind of defense.
>
> Thank you again for your engagement and suggestions. We would welcome any final feedback or comments!

---

> > ### Author Response · Authors · 2024-12-03
> > **Note on BEAST perplexity**
> >
> > We should also note that our new attack, BEAST, is specifically designed to sample low-perplexity adversarial tokens.
> >
> > From Section 3.2 of https://arxiv.org/pdf/2402.15570:
> >
> > > In our case, BEAST maintains the readability by multinomially sampling the adversarial tokens based on the target LM’s next token probability distribution.
> >
> > In Table 3, Figure 6 and Table 7, they find that BEAST is still able to attack successfully most of the time even in the presence of a perplexity-filtering defense (PPL), as long as the beam size is not too large. Indeed, we used a beam width of 7 in our evaluations which holds up well against this defense.

---

### Official Review · Reviewer_zMEX · 2024-11-03

**Soundness:** 3
**Presentation:** 3
**Contribution:** 1
**Rating:** 3
**Confidence:** 3

**Summary:**

The paper examines how model robustness to adversarial attacks and jailbreaks scales with model size. To investigate this, the authors evaluate various Pythia models fine-tuned on six binary classification datasets, such as for automatic spam detection. They use random tokens and greedy coordinate gradient as attack methods. For standard models, the authors observe that model scaling generally improves robustness, though results vary by task, and even the largest models are not fully robust. The authors also evaluate adversarially trained models, finding that these models are more robust, and assess their robustness against stronger and unseen attacks. Finally, they analyze the balance between offensive and defensive robustness.

**Strengths:**

- Overall, the claims in the paper are backed up by sufficient evidence for the configuration of models/attacks presented in the paper.
- The results are clearly presented. Plot layout and writing style are fine.

**Weaknesses:**

- The focus on classification models feels restrictive. Modern LLMs are primarily used as generative models. While focusing on binary classification is reasonable for ease of evaluation, I believe the paper would benefit from some evaluation in a generative setting to see if the findings hold.
-  My largest complaint is about the contributions of this paper. Many findings in this paper are not particularly surprising. The observation that model size improves robustness to a point is in line with previous work (e.g., Ganguli et al., 2022). Similarly, adversarial training enhancing robustness is well-known. Transfer protection against other attacks has been frequently evaluated for image classification models, so it’s somewhat expected to see similar trends in LLMs.
- Evaluating results on a single family of models with different sizes (like Pythia) is sensible for this study. However, it would also be valuable to test transferability to various pre-trained LLMs from other model families, such as LLaMA, Mistral, Vicuna, etc.
- There’s insufficient evaluation of recent attacks. For a study like this, a broader evaluation of attacks would strengthen the findings. Some relevant recent works include:

Andriushchenko, M., Croce, F., & Flammarion, N. (2024). "Jailbreaking Leading Safety-Aligned LLMs with Simple Adaptive Attacks." arXiv preprint arXiv:2404.02151.

Liu, X., Xu, N., Chen, M., & Xiao, C. (2023). "AutoDAN: Generating Stealthy Jailbreak Prompts on Aligned Large Language Models." arXiv preprint arXiv:2310.04451.

Sadasivan, V. S., Saha, S., Sriramanan, G., Kattakinda, P., Chegini, A., & Feizi, S. (2024). "Fast Adversarial Attacks on Language Models In One GPU Minute." arXiv preprint arXiv:2402.15570.

Chao, P., Robey, A., Dobriban, E., Hassani, H., Pappas, G. J., & Wong, E. (2023). "Jailbreaking Black Box Large Language Models in Twenty Queries." arXiv preprint arXiv:2310.08419.

**Questions:**

In Figure 3, the slopes of some graphs corresponding to larger models with lower ASR don’t appear to completely flatten out. Do the authors believe these large models have a certain level of inherent robustness, or is the low success rate simply an artifact of insufficient compute, with all models potentially reaching 100% ASR with enough resources?

---

> ### Author Response · Authors · 2024-11-16
>
> Thank you for your helpful comments. We have written out a [response to all reviewers](https://openreview.net/forum?id=IAFLoDz6H5&noteId=GK46muABVY) and [plan for the rebuttal period](https://openreview.net/forum?id=IAFLoDz6H5&noteId=w0Lg16u4fn), and would welcome any feedback. We respond to your specific questions and comments below.
>
> > The focus on classification models feels restrictive. Modern LLMs are primarily used as generative models. While focusing on binary classification is reasonable for ease of evaluation, I believe the paper would benefit from some evaluation in a generative setting to see if the findings hold.
>
> We agree that it would be beneficial to verify that the trends hold in a generative setting. To this end, we are testing the adversarial robustness of Qwen on the StrongREJECT task during the rebuttal period and will include it in the paper as results come in.
>
> > My largest complaint is about the contributions of this paper. Many findings in this paper are not particularly surprising. The observation that model size improves robustness to a point is in line with previous work (e.g., Ganguli et al., 2022).
>
> * We assume you are referring to https://arxiv.org/pdf/2209.07858, please let us know if you are thinking of a different paper.
> * It’s true that Figure 1 in that paper shows robustness of different models across three model scales (2.7B, 13B, 52B). However:
>     * The authors only show 3 models spanning ~1.5 orders of magnitude; we study 10 models across 3 orders of magnitude
>     * The Figure does not show particularly salient trends in the effect of scale on robustness. For example, the RL LM gets better at 13B and worse at 52B, and the Prompted LM appears to steadily get worse with scale.
>     * The authors use a fixed-strength attack across model sizes and do not quantify the effects of increasing attack compute.
>     * Our study is significantly more systematic in that we study scaling across multiple seeds and tasks, and consider model size, attack compute, and defense compute (adversarial training) in the calculation.
> * We agree that the finding that model size generally improves robustness is not surprising in itself. However, our analysis goes further than this, showing the magnitude (and not just direction) of the effect, the existence of a task which contradicts the prevailing trend, and scaling curves for attack compute.
>
> > Similarly, adversarial training enhancing robustness is well-known. Transfer protection against other attacks has been frequently evaluated for image classification models, so it’s somewhat expected to see similar trends in LLMs.
>
> As you suggest, it is not surprising that adversarial training improves model robustness for language models. But our analysis does not stop there. We plot the attack success rate curves over the course of adversarial training, finding a roughly linear trend (given a logit transformation on the y axis) between compute spend and 1 - attack success rate, independent of model size. We also find the larger models are better able to generalize their robustness, and that the offense/defense balance favors the attacker even as we scale model size.
>
> > Evaluating results on a single family of models with different sizes (like Pythia) is sensible for this study. However, it would also be valuable to test transferability to various pre-trained LLMs from other model families, such as LLaMA, Mistral, Vicuna, etc.
>
> We agree. We are currently training Qwen on our tasks and aim to have results out by the end of the rebuttal period. Is there a specific model family you’d like to see in this paper?
>
> > There’s insufficient evaluation of recent attacks. For a study like this, a broader evaluation of attacks would strengthen the findings.
>
> We agree it would be good to have at least one more attack to ensure that our results are generally applicable, and we are planning to implement BEAST as a state-of-the-art black-box attack. Does this address your concern?
>
> > In Figure 3, the slopes of some graphs corresponding to larger models with lower ASR don’t appear to completely flatten out. Do the authors believe these large models have a certain level of inherent robustness, or is the low success rate simply an artifact of insufficient compute, with all models potentially reaching 100% ASR with enough resources?
>
> We suspect it’s the latter: that running GCG for long enough would lead to 100% ASR eventually. However, this might take more compute than the pretraining compute, so it might be infeasible to get to that ASR in practice. Indeed, one of our future work directions is to more carefully quantify the absolute costs as well as relative costs in our exploration of the offense-defense balance.

---

> ### Author Response · Authors · 2024-12-02
> **Following up on new results**
>
> We have updated our paper with new experimental results—see [here](https://openreview.net/forum?id=IAFLoDz6H5&noteId=9sGAdINuRf) for a full description.
>
> We updated the paper to more clearly motivate our design decisions and performed several additional experiments.
> Specifically, we reproduced our experiments on the [Qwen2.5](https://github.com/QwenLM/Qwen2.5) Base model family, we adversarially attacked Qwen2.5 Instruct on a generative task ([StrongREJECT](https://arxiv.org/pdf/2402.10260])), and we implemented a new state-of-the-art black box attack ([BEAST](https://arxiv.org/pdf/2402.15570)).
>
> We hope that these results address your concerns about whether our results generalize to a modern model family (they mostly do), a new attack (they do), and a generative experiment (they do).
>
> Please let us know if you have additional comments or suggestions. Thank you.

---

### Official Review · Reviewer_EWYP · 2024-11-03

**Soundness:** 3
**Presentation:** 4
**Contribution:** 3
**Rating:** 8
**Confidence:** 4

**Summary:**

The paper investigates the impact of scale on the adversarial robustness of language models, exploring if larger models inherently resist attacks better than smaller ones. Through empirical testing on binary text classification datasets, it finds that while larger models show some improved robustness, this effect is inconsistent across tasks. The study also examines the efficiency of adversarial training in enhancing robustness, observing that it is far more compute-efficient than scaling alone. Finally, the paper explores the offense-defense balance, revealing that adversarial training alone may not always suffice for robustness, especially as models scale.

**Strengths:**

The paper is a valuable empirical contribution to understanding the limitations and benefits of scaling and adversarial training for LLM robustness:
- The study covers all model sizes from the Pythia scaling suite, and a variety of binary text classification tasks (IMBD, Spam, Harmful and Harmless, PasswordMatch, WordLength), providing a broad analysis of robustness trends across different scenarios.
- The paper’s analysis on adversarial training and its comparison to scaling as a defense mechanism is interesting and insightful.
- The authors perform many further analyses, for example on the defense vs offense compute balance / trade off, as well as the transferability of adversarial training, which may be crucial for the future of LLM safety.

**Weaknesses:**

I believe the main weaknesses are along the line of “completing the picture”; e.g. performing further experiments to get a more complete picture of LLM robustness scaling:
- Some findings appear to be task-dependent, making it hard to draw a general conclusion. Could we perform further ablation studies to understand why an approach is more successful on some tasks against some others?
- The authors use two main adversarial attacks (RandomToken and GCG) that are used extensively in evaluations, but one could perhaps explore further attack methods, for example (https://arxiv.org/abs/2404.02151).
- Another thing that I believe would need investigation is whether robustness could be an emergent property (Wei et al., 2022, Schaeffer et al., 2023), e.g. if models suddenly become (more) robust after some particular scale. Exploring these with open models going beyond the Pythia suit would add further important insights.
- Finally, the paper focuses only on classification tasks, but the typical use of LLMs is generative. Doing similar analyses on the generative side, using standard jailbreaking benchmarks and techniques, and seeing if the paper’s findings transfer there would be also crucial.

**Questions:**

My questions are mostly towards addressing some of the weaknesses mentioned:
- Could the authors clarify the reasoning behind the choice of specific tasks (e.g., IMDB, Spam)? How do these tasks represent the broader applicability of their findings?
- Can the authors explain why adversarial training shows significant effectiveness in some tasks but not in others (e.g., WordLength task)? Is it possible to perform some ablation studies to better understand the reasons behind the differences in effectiveness, and what are the general takeaways?
- Could it be the case that robustness is an emergent property, and is it possible to investigate this in the context of the paper?
- Would the authors consider extending their analysis to include generative models? Given that LLMs are often used for generation rather than classification, robustness trends might differ in such contexts.

---

> ### Author Response · Authors · 2024-11-16
>
> Thank you for your helpful comments. We have written out a [response to all reviewers](https://openreview.net/forum?id=IAFLoDz6H5&noteId=GK46muABVY) and [plan for the rebuttal period](https://openreview.net/forum?id=IAFLoDz6H5&noteId=w0Lg16u4fn), and would welcome any feedback. We respond to your specific questions and comments below.
>
> > I believe the main weaknesses are along the line of “completing the picture”; e.g. performing further experiments to get a more complete picture of LLM robustness scaling
>
> We agree!
>
> > Some findings appear to be task-dependent, making it hard to draw a general conclusion. Could we perform further ablation studies to understand why an approach is more successful on some tasks against some others?
> > Can the authors explain why adversarial training shows significant effectiveness in some tasks but not in others (e.g., WordLength task)? Is it possible to perform some ablation studies to better understand the reasons behind the differences in effectiveness, and what are the general takeaways?
>
> It’s an interesting question. We were surprised by the WordLength results too and left them in for the purposes of transparency. Our initial guess was that it had to do with each of the words usually being a single token and thus being more of a memorization task than an understanding/reasoning task. However, pre-attack accuracies on the task are very high, and we have no leakage from training dataset to validation dataset, so this would appear to debunk that understanding.
>
> Is there any particular ablation you’d like to see on WordLength or any other task?
>
> > The authors use two main adversarial attacks (RandomToken and GCG) that are used extensively in evaluations, but one could perhaps explore further attack methods, for example (https://arxiv.org/abs/2404.02151).
>
> We are adding BEAST (https://arxiv.org/abs/2402.15570) as an additional attack method and hope to have results for it by the end of the rebuttal period.
>
> The specific attack you link to is in fact similar to our RandomToken attack, with the addition of custom prompts designed for each attacked model. Since we want to study general scaling trends, we have aimed to focus on attacks that do not require customization to a given model family.
>
> > Another thing that I believe would need investigation is whether robustness could be an emergent property (Wei et al., 2022, Schaeffer et al., 2023), e.g. if models suddenly become (more) robust after some particular scale.
> > Could it be the case that robustness is an emergent property, and is it possible to investigate this in the context of the paper?
>
> This is a very interesting question, and indeed was something we were trying to investigate. Over our model scales, we found that adversarial robustness did not appear to emerge as model scale increased, however there are emergence-like phenomena in the sample efficiency of models from small amounts of adversarial training (more specifically, for the leftmost datapoints in Figures 6 and 8 there appears to be a jump in how much robustness a model can get out of the first 200 adversarial training examples, with around 100M parameters being the transition point).
>
>
> [continued in next comment]

---

> ### Author Response · Authors · 2024-11-16
>
> [continued from previous comment]
>
> > Exploring these with open models going beyond the Pythia suit would add further important insights.
>
> We are running the same experiments on the Qwen family and intend to have the results included in the paper by the end of the rebuttal period.
>
> > Finally, the paper focuses only on classification tasks, but the typical use of LLMs is generative. Doing similar analyses on the generative side, using standard jailbreaking benchmarks and techniques, and seeing if the paper’s findings transfer there would be also crucial.
> > Would the authors consider extending their analysis to include generative models? Given that LLMs are often used for generation rather than classification, robustness trends might differ in such contexts.
>
> We agree that it would be very interesting to verify that our results generalize to the generative setting. We are running a generative experiment on Qwen (the StrongREJECT task) which we intend to include in the paper before the end of rebuttal period
>
> > Could the authors clarify the reasoning behind the choice of specific tasks (e.g., IMDB, Spam)? How do these tasks represent the broader applicability of their findings?
>
> Sure! In general, we were looking for tasks that covered a spread of settings (natural language, algorithmic, preference modelling) and were real-world relevant.
> * IMDB, Spam were chosen as standard, real-world-applicable, not-too-difficult natural language understanding tasks (sentiment analysis and spam detection).
> * PasswordMatch and WordLength were created as more “algorithmic” tasks, which are less about using natural language understanding from pretraining and more about “reasoning” about the text that is provided in the prompt. They are quite easy tasks.
> We wanted to focus on tasks which can be accomplished to a high degree of accuracy by a range of model sizes, in order to avoid conflating the attack being strong with the model struggling to do the task and being “lightly pushed over the decision boundary” by the attack (granted, we did not study the decision boundary geometry explicitly).
> * Helpful and Harmless are quite challenging preference datasets presented in (Bai et al., 2022) and used for RLHF.
>
> We suspect that the findings on these tasks will generalize to other classification settings. It remains to be determined to what extent they generalize to generative settings, but initial results have pointed in the generalization direction.

---

> ### Comment · Reviewer_EWYP · 2024-11-25
>
> Thank you very much for your detailed response!
> - Your reasoning about the dataset selection makes now sense to me.
> - For the ablations, my only concern is that the trends observed are often noisy, and depend a lot on the particular task; ideally, I'd like to understand a bit better why this happens - is it an issue of prompting, the PILE training set, hyper-parameters? Does it happen on other models too, except the Pythia family? But I also acknowledge that figuring this out can be challenging.
>
> Overall, the authors' response addresses most of my concerns, and I fully agree with the rebuttal plan; I will look at the new results once they're released.

---

> ### Author Response · Authors · 2024-12-02
> **Following up on new results**
>
> Thank you for your response! We have [updated the paper](https://openreview.net/pdf?id=IAFLoDz6H5) and written a [description of the new results](https://openreview.net/forum?id=IAFLoDz6H5&noteId=9sGAdINuRf). Please let us know if you have any additional comments or suggestions!
>
> > For the ablations, my only concern is that the trends observed are often noisy, and depend a lot on the particular task; ideally, I'd like to understand a bit better why this happens - is it an issue of prompting, the PILE training set, hyper-parameters? Does it happen on other models too, except the Pythia family? But I also acknowledge that figuring this out can be challenging.
>
> As you say, it is an interesting and challenging question. For the finetuned tasks (Figure 2, 10, 11, 12), our best guess as to why we observed so much variability across attack success rate when attacking different finetuning seeds of the same model sizes is the following: the pretraining and finetuning distributions do not contain adversarial examples (or any safety training), and thus whether or not a finetuned model is robust to adversarial attack is largely up to chance—in other words, where in parameter space the model ends up after finetuning (with some parts of parameter space being naturally more robust than others). We observe this variability in both the Pythia models and in the Qwen2.5 Base models in our new experiments, which suggests that this phenomenon is not unique to the Pile or the Pythia architecture.
>
> We observe less variability across models that have had some kind of safety training. For one thing, when we adversarially train models, their robustness becomes more predictable, and improves relatively smoothly as a function of defense compute, regardless of what robustness they started from after finetuning. Additionally, in our new experiment on Qwen2.5 Instruct (which based on the Qwen2 technical report we strongly suspect was safety-trained), we observe a clean trend of improvement in robustness across model sizes (since we did not finetune this model we could not use different finetuning seeds, and instead reported the 95% Wilson Score Interval as a measure of spread).
>
> It is worth noting that we also observe clean trends in attack scaling on both base models and safety-trained models. So while finetuned-only models show significant variability in robustness across finetuning seeds, from the attacker's perspective, the additional compute needed to _increase_ attack success rate by a given amount scales very smoothly, regardless of the _starting_ robustness of a given seed.
>
> Does this help answer your question?

---

> > ### Comment · Reviewer_EWYP · 2024-12-02
> >
> > - Yes, this addresses my question on the task variability, it's on the directions that I was thinking too. The observation that the safety training has an important influence on it is also interesting.
> > - Moreover, the finding about the relationship between the attack success increment and compute is important, perhaps it could be shown also more clearly in the paper with additional plots.
> > - This and the updated paper cover all my concerns, I'd like to thank again the authors for their efforts and engagement!

---

> ### Author Response · Authors · 2024-12-03
>
> Thank you for your thoughtful comments throughout the review process! We are happy to hear that our current revision and comments addressed all of your concerns.
>
> For the final version of the paper, we are also planning to run experiments on Llama 3, and look forward to seeing whether the variability across seeds on those models is similar to Pythia and Qwen 2.5.
>
> Regarding the relationship between the attack success increment and compute, is there any particular plot that you think would be useful to add?

---

> > ### Comment · Reviewer_EWYP · 2024-12-03
> >
> > No worries, I was just thinking, if we overlay plots of the type attack success increment vs compute (relative to pre-training) for different tasks (for the same model family and attack type), whether we're going to observe any regularities. But judging qualitatively from the other graphs, it seems that the slopes for different tasks will be different, so I'm not sure if such a plot is really needed.

---

> ### Author Response · Authors · 2024-12-03
>
> That is also an interesting question! Appendices C.5.2 and C.5.3 show the slopes of the GCG and RandomToken attacks. As you mention, qualitatively it does look like the slopes are different across tasks in general. That said, for GCG, with reference to Figures 13, 14, 15, we do notice similarities between tasks which are “of the same type”. Spam and IMDB, our “real world” datasets, look fairly similar (initially constant slopes with a “jump” at some point—70M->160M for Spam, and 1.4M->2.6B for IMDB). PasswordMatch and WordLength, our algorithmic tasks, are also similar (fairly constant slopes across model size). Helpful and Harmless, the RLHF datasets, look very similar (clean and predictably changing slopes except for the smallest Pythia model). The RandomToken slopes (Figures 16, 17, 18) show similar behavior except in the case of Spam and IMDB, where the slopes are cleaner (Spam) and less clean (IMDB) than the GCG equivalents respectively. Overall, the more realistic tasks are more challenging, as seen by the lower pre-attack accuracy: one explanation for the observed slopes is that the attacks would be particularly successful against small models which struggle the most with these tasks, and larger models benefit from their improved capabilities, resulting in a greater change in attack scaling across model sizes.
>
> As we obtain results with more sizes, datasets and seeds using Qwen2.5, it will be interesting to see which of these details remain true for each dataset, and we will be sure to include commentary highlighting this in the camera-ready version of the paper. For Pythia, the slope plots are quite jumpy, and it will be good to compare whether Qwen2.5 exhibits similar transitions at certain size thresholds.

---

> ### Author Response · Authors · 2024-12-04
> **Overlayed plots**
>
> In line with your suggestion to overlay the plots for attack scaling across tasks, we link here to overlayed regression lines for attack scaling vs. model size across tasks for the [GCG attack](https://pasteboard.co/cyC528MHkwgD.png) and [RandomToken attack](https://pasteboard.co/eMqJEYMTzZCm.png). As you suggested it might, viewing these attacks on the same axes highlights similarities and differences across attacks (some of which we mentioned in the [previous comment](https://openreview.net/forum?id=IAFLoDz6H5&noteId=pH0NeKEryk)). For most tasks (Spam, IMDB, Helpful, Harmless), larger models show worse returns for the attacker as they scale up their compute. In contrast, the algorithmic datasets (PasswordMatch and WordLength) show a flat or even positive trend for the attacker with victim model scale. We also note that while RandomToken exhibits much worse attack scaling than GCG in absolute terms (to be expected since it is a much weaker attack), the trends across tasks are broadly similar between the two attacks.

---

### Official Review · Reviewer_4pZx · 2024-11-03

**Soundness:** 2
**Presentation:** 3
**Contribution:** 2
**Rating:** 3
**Confidence:** 4

**Summary:**

The authors conduct an empirical investigation into scaling trends for the adversarial robustness of language models. With some toy settings, the study indicates that
- larger models are generally more resistant to attacks, with variability across tasks;
- adversarial training boosts robustness for all model sizes, and scaling adversarial training is cost-effective compared to pre-training;
- adversarial training against one attack transfers protection to similar attacks;
- the offense/defense balance varies by task and model size.

**Strengths:**

- The topic is generally interesting and could be beneficial to the community.
- Some conclusions, if also held for other LLMs, are interesting.

**Weaknesses:**

- The experiment setting is problematic.
- - Only toy tasks and language models are used. Instead of evaluating LLMs in a few-shot/zero-shot way, the paper fine-tunes the Pythia family of models on some classification tasks. Pythia models are only pre-trained on general corpus without any RLHF and their performances are well-known to be far lower than LLMs like LlaMa and Gemma. I doubt how largely the results in the paper can be generalized to other LLMs.
- - The attack methods are naive. Two attack methods are considered. One is to randomly add some tokens as suffixes of the inputs, while the other one generates a universal adversarial suffix. Since the paper is only considering the toy setting with classification tasks, I don't see any reason why other classical attack methods in NLP are not used. For example, check the following papers:
- - - [1] Jin, Di et al. “Is BERT Really Robust? A Strong Baseline for Natural Language Attack on Text Classification and Entailment.” AAAI Conference on Artificial Intelligence (2019).
- - - [2] Hou, Bairu et al. “TextGrad: Advancing Robustness Evaluation in NLP by Gradient-Driven Optimization.” ArXiv abs/2212.09254 (2022): n. pag.
- - - [3] Li, Linyang et al. “BERT-ATTACK: Adversarial Attack against BERT Using BERT.” ArXiv abs/2004.09984 (2020): n. pag.


Since this is largely an empirical paper, I would like to encourage the authors to refine the setting and conduct more comprehensive experiments to fulfill the goal.

**Questions:**

Please refer to those in the weaknesses section.

---

> ### Author Response · Authors · 2024-11-16
>
> Thank you for your helpful comments. We have written out a [response to all reviewers](https://openreview.net/forum?id=IAFLoDz6H5&noteId=GK46muABVY) and [plan for the rebuttal period](https://openreview.net/forum?id=IAFLoDz6H5&noteId=w0Lg16u4fn), and would welcome any feedback. We respond to your specific questions and comments below.
>
> > The experiment setting is problematic.
> > Only toy tasks and language models are used. Instead of evaluating LLMs in a few-shot/zero-shot way, the paper fine-tunes the Pythia family of models on some classification tasks.
>
> The reason we decided to do Pythia fine-tuning was because it was the only way we could see of studying the scaling behavior across 3 orders of magnitude (we estimate we can get a maximum of ~1.8 orders of magnitude out of generative models since in our experience generative models below 1B parameters are either incoherent or trivially jailbroken). Since we are focused on identifying scaling trends, this was a useful setting for us. See the [response to all reviewers](https://openreview.net/forum?id=IAFLoDz6H5&noteId=GK46muABVY) for more details.
>
> During the rebuttal period, we plan to run a generation (not classification) robustness experiment on Qwen on the StrongREJECT task.
> Is there any specific task that you would like us to include in our analysis? We agree that PasswordMatch and WordLength are toy tasks—we intentionally chose them to have a more “algorithmic” flavor; Spam and IMDB are relatively easy but still real-world-relevant; Helpful and Harmless are challenging tasks (finetuned models get ~70% performance on clean data) used for RLHF.
>
> > Pythia models are only pre-trained on general corpus without any RLHF and their performances are well-known to be far lower than LLMs like LlaMa and Gemma. I doubt how largely the results in the paper can be generalized to other LLMs.
>
> We agree that investigating state of the art generative models is a compelling next step. During the rebuttal period, we plan to reproduce our experiments on Qwen in the classification setting to ensure that our results are not Pythia-specific.
>
> > The attack methods are naive. Two attack methods are considered. One is to randomly add some tokens as suffixes of the inputs, while the other one generates a universal adversarial suffix. Since the paper is only considering the toy setting with classification tasks, I don't see any reason why other classical attack methods in NLP are not used.
>
> We decided to focus on GCG with an adversarial suffix because it is considered a standard state-of-the-art attack, commonly used against autoregressive language models. Note that our version of GCG is not universal: each adversarial suffix is found for a specific datapoint, not all datapoints. Because of this, it is in fact a very strong attack.
>
> Since we were interested primarily in the existence of vulnerabilities as model size increases we chose to focus on a state-of-the-art GCG white-box attack and a black-box RandomToken baseline. That said, we agree that our results would be stronger if we verified them with another state-of-the-art attack, and we are hoping to have results for BEAST (a state-of-the-art black-box attack) before the end of the rebuttal period.
>
> > Since this is largely an empirical paper, I would like to encourage the authors to refine the setting and conduct more comprehensive experiments to fulfill the goal.
>
> Thank you for your comments and suggestions. Do our [proposed experimental plan](https://openreview.net/forum?id=IAFLoDz6H5&noteId=w0Lg16u4fn) and responses to your questions address your concerns?

---

> ### Author Response · Authors · 2024-12-02
> **Following up on new results**
>
> We have updated our paper with new experimental results—see [here](https://openreview.net/forum?id=IAFLoDz6H5&noteId=9sGAdINuRf) for a full description.
>
> Along with updating the paper to more clearly motivate our design decisions, we performed several additional experiments.
> Specifically, we reproduced our experiments on the Qwen2.5 Base model family, we adversarially attacked Qwen2.5 Instruct on a generative task ([StrongREJECT](https://arxiv.org/pdf/2402.10260)), and we implemented a new state-of-the-art black box attack ([BEAST](https://arxiv.org/pdf/2402.15570)).
>
> We hope that these additions address your concerns about the relevance of the experimental setting and the strength of the attacks considered.
>
> Please let us know if you have additional comments or suggestions. Thank you.

---

### Official Review · Reviewer_Vfij · 2024-11-04

**Soundness:** 3
**Presentation:** 3
**Contribution:** 2
**Rating:** 6
**Confidence:** 3

**Summary:**

This paper explores whether scaling language model size improves robustness against adversarial attacks, focusing on binary classification tasks. It finds that larger models show modestly increased robustness, though with variability across tasks. Adversarial training, rather than model scaling alone, is found to be more effective for enhancing robustness across various threat models.

**Strengths:**

- This paper addresses a crucial gap in understanding the effect of model scaling on language model robustness through empirical evaluations.
- It provides a decent empirical analysis across multiple model sizes, attack types, and adversarial training setups.
- It identifies that adversarial training is more compute-efficient than model scaling for achieving robustness, adding practical insight.
- This paper highlights task-specific trends in robustness, which is valuable for real-world application considerations.

**Weaknesses:**

- The study mostly focus on pythia, a decoder only LM. It is hard to convince if the same observation could be made by other structures such as T5 and some encoder only models.
- The study also didn't account if the conclusion hold consistent across other decoder only models.
- The paper limits its experiments to relatively straightforward binary classification tasks like spam detection, and sentiment analysis on the IMDB dataset. While these provide useful insights, they may not fully test robustness in complex or real-world applications, such as multi-label or sequential tasks that require nuanced understanding and context retention.
- Some other adversarial training such as ALUM is not discussed in the experiment. The findings may not be extended to all sorts of adversarial training.

**Questions:**

- Are this findings invariant to model structures? Why pythia over other open source models?
- How does the quality and diversity of pretraining data influence model robustness?
- Since this paper focuses on classification tasks, what insights (if any) does it provide for generative models, which may face different types of adversarial attacks?
- Are the conclusions on adversarial trainings hold consistent over all variations of such trainings such as token level, augmentation, PGD? Some trainings may cost large amount of time and resources to train.

---

> ### Author Response · Authors · 2024-11-16
>
> Thank you for your helpful comments. We have written out [a plan](https://openreview.net/forum?id=IAFLoDz6H5&noteId=w0Lg16u4fn) for what we will do during the rebuttal period and would welcome any feedback on that or [our overall comment](https://openreview.net/forum?id=IAFLoDz6H5&noteId=GK46muABVY).
>
> > The study also didn't account if the conclusion hold consistent across other decoder only models.
>
> We are expanding our study to include Qwen in order to verify that the results hold across another decoder-only model family.
>
> > The study mostly focus on pythia, a decoder only LM. It is hard to convince if the same observation could be made by other structures such as T5 and some encoder only models.
>
> We agree that the experimental results would be more exhaustive if we also studied encoder/decoder and encoder-only models. We chose to focus on decoder-only models since this seems to be the paradigm that frontier models have converged on.
>
> > The paper limits its experiments to relatively straightforward binary classification tasks like spam detection, and sentiment analysis on the IMDB dataset. While these provide useful insights, they may not fully test robustness in complex or real-world applications, such as multi-label or sequential tasks that require nuanced understanding and context retention.
>
> It’s true that our setting fails to capture some of the real-world complexity present in modern chat systems. We chose the binary classification setting for several reasons, including availability of performant models across 3 orders of magnitude and simple definition of “attack success”—see the response to all reviewers. As you mention, multi-label and especially sequential tasks are strictly harder than the binary classification setting, so we can consider attack success rate on binary classification akin to an upper bound for robustness of these other settings with larger attack surface area.
>
> > Some other adversarial training such as ALUM is not discussed in the experiment. The findings may not be extended to all sorts of adversarial training.
>
> Other methods of adversarial training may have other results, and we would be excited about future research which explores differences in scaling trends when using adversarial training in the embedding space (such as ALUM). Unfortunately, analyzing a variety of adversarial training methods would have been prohibitively expensive for us. Since we were focused primarily on understanding general scaling trends, including the offense/defense balance of scaling compute, we chose to focus on the straightforward apples-to-apples comparison of using the same attacks for both adversarial attacks and training. We will modify the paper to make this focus more explicit.
>
> > Are this findings invariant to model structures? Why pythia over other open source models?
>
> See response to all reviewers.
>
> > How does the quality and diversity of pretraining data influence model robustness?
>
> This is a very interesting question! In our initial experiments, we noticed a large amount of variability in out-of-the-box robustness which appeared to vary randomly based on pretraining checkpoint. This is part of the reason we moved to the adversarial training setting: results in the finetuned-only setting varied so much based on what we assume was pretraining randomness. Beyond this, we have not looked at the effects of pretraining. This is for two reasons:
> 1) It would be prohibitively expensive for us to perform pretraining on the larger models.
> 2) The standard paradigm for large language models appears to be pretraining first, and focusing on safety-tuning after pretraining is completed. We wanted our study to be as applicable as possible to what is currently done in frontier AI labs.
>
> > Since this paper focuses on classification tasks, what insights (if any) does it provide for generative models, which may face different types of adversarial attacks?
>
> We chose GCG precisely because this is a common attack used against generative models. We suspect similar results for the generative setting, and are hoping to have corroborating Qwen results before the end of the rebuttal period.
>
> > Are the conclusions on adversarial trainings hold consistent over all variations of such trainings such as token level, augmentation, PGD? ...
>
> This is a good question. We chose simple adversarial training as a “standard” technique for making models more robust. It is indeed quite expensive to train for the larger models. We tried both a black-box attack (RandomToken) and a state-of-the-art token-level white-box attack (GCG). Adversarial training on both was successful, though adversarially training on RandomToken did not transfer to defending against the much stronger GCG. As a strong attack, we suspect GCG would compare favorably against augmentation and PGD. We suspect latent-space attacks would be less expensive to adversarially train on than token-level ones, though we have not included them in this work.

---

> > ### Comment · Reviewer_Vfij · 2024-11-16
> >
> > Thank you for the response! I agree that studying the effect of pretraining data and some other adversarial trainings would be very expensive to scale up. Most of my concerns have been addressed. I will consider to adjust the score once I have fully checked the new results.

---

> ### Author Response · Authors · 2024-12-02
> **Following up on new results**
>
> Thank you for the updated score! Please let us know if you have any additional comments or suggestions.

---

### Author Response · Authors · 2024-11-16
**Response to all reviewers**

Thank you to all our reviewers for the helpful comments. Thanks to your feedback, we realized that we were not clear enough with the motivation and framing of our paper, and will address this in a revision which we will upload in the coming days. We will present some clarifying points here as well.

Our main clarification is that this paper focuses specifically on understanding the effects of scaling on adversarial robustness, rather than trying to use scaling as a way to make models adversarially robust. The current generation of language models are not robust to adversarial attack. How will increasing compute and model size affect the robustness of future models? Additional compute helps adversarial training, but what if the attacker also scales up their compute? Importantly, answering these questions does not require a state of the art model, attack, or defense; what is important is to select these elements such that the results will plausibly generalize to future frontier models.

Our main conclusions are as follows:
1) Simply increasing model size confers a minimal but measurable improvement in robustness.
2) All models benefit from adversarial training and can use it to generalize to a stronger version of the same attack they were trained on. Large models are more sample efficient, and only large models are able to generalize their defense against a different threat model. Thus, scaling is not a panacea, it is rather a multiplier and a generalizer for specific interventions (like adversarial training) that are performed on the model.
3) Defense compute scaling for adversarial training smoothly improves adversarial robustness. Similarly, attack compute scaling smoothly improves attack performance for both our black-box (RandomToken) and our white-box (GCG) attacks, against both finetuned and adversarially trained models.
4) As attacker and defender both scale up compute, the attacker maintains an advantage. As the defender doubles the compute spent on adversarial training, the attacker needs to less than double the compute spent on adversarial attacks to maintain the same attack success rate. This effect is not mitigated by increasing model size.

We also take this opportunity to respond to concerns shared by multiple reviewers. Specifically, several reviewers were concerned by our choice of the classification setting rather than a more realistic generative “jailbreak” setting, and by our choice of the Pythia model suite. Both of these choices were intentional in enabling us to systematically study scaling without the compute of a large industry lab.

## Choice of classification setting
While instruction-following models can be trained at smaller model sizes, models are not typically good at refusal until above the ~1B parameter range. Llama 405b does not fit on an 8xH100 node without quantization below 16 bits, so the largest usable models without significant performance-relevant implementation differences or extensive engineering efforts cap out around 70B parameters, resulting in less than 2 orders of magnitude of variation, making it difficult to measure the effects of scaling across a wide range of model sizes. In contrast, classification models are able to attain high performance at very small model scales, allowing us to study the variation in model robustness over 3 orders of magnitude of model size within a single model family.

Binary classification also allows us to unambiguously study the success of an attack. Many jailbreaks get the model to “comply” without providing useful information (https://arxiv.org/abs/2402.10260), and models can seemingly reject a request which they then go on to fulfill.

Classification is also relevant in the real-world, used in important practical tasks such as spam detection, antifraud, and reward modeling. Indeed, we can think of binary classification as an upper bound on the robustness of a generative model, since classification is strictly an easier task than generation: any LLM which rejects some requests and answers others is implicitly classifying prompts as inputs to reject or fulfill.

## Choice of the Pythia model suite
As mentioned in the previous subsection, it is crucial for our experiment to study robustness behavior across multiple orders of magnitude, within the same model family. When we started the project, while more modern models like Llama 2 were available, Pythia was the most modern autoregressive language model suite that provided 10 model sizes across 3 orders of magnitude. Over the course of the project, other model families such as Qwen (https://github.com/QwenLM/Qwen2.5) and Spectra (https://arxiv.org/abs/2407.12327) have been made available, and we are currently working to bring our results to Qwen as well (see the rebuttal plan).

---

### Author Response · Authors · 2024-11-16
**Rebuttal period plan**

We plan to make the following modifications and additional experiments in response to the reviews.
* Improve the motivation and framing of our experiments in the paper
* Run our classification experiments using the Qwen2.5 model family on the Spam and Harmless tasks to ensure that our results generalize to a more modern model family
* (stretch) Benchmark the Qwen2.5 model family on StrongReject, to confirm that our results hold in the generative setting
* (stretch) Evaluate our models with an additional adversarial attack (likely BEAST), to confirm that our results hold for a state-of-the-art black-box attack

We aim to complete these four tasks during the rebuttal period, though are less confident in our ability to do the last two in this timeframe, and might only have results after the end of the period. We will keep the reviewers updated on our progress.

---

### Author Response · Authors · 2024-11-23
**Updated paper draft; new experiments forthcoming**

As per our [rebuttal period plan](https://openreview.net/forum?id=IAFLoDz6H5&noteId=w0Lg16u4fn), we have updated the paper to improve motivation and framing for the experiments completed prior to submission.

The new experiments described in the plan are currently being run. We plan to include the new results in a further updated draft early next week.

---

### Author Response · Authors · 2024-11-28
**Updated paper with new experiments**

We have updated the paper to include our new experimental results.

We will follow up here with a description of the new experiments by EoD Friday.

---

### Author Response · Authors · 2024-11-30
**Description of our new results (1/2)**

We are pleased to present the reviewers with our new results, which can be seen in our [updated paper](https://openreview.net/pdf?id=IAFLoDz6H5).

We present our new results with reference to our rebuttal period plan, which we go through below.

*TL;DR: we evaluate our results on a new model family (Qwen2.5), implement a new attack (BEAST), and explore a generative task (StrongREJECT). Our results with the new attack closely match our results with GCG/RandomToken, and our results on the generative task (with a model that has undergone instruction finetuning) show similar behavior as the classification models after some adversarial training. Our results on the new Qwen2.5 model family mostly match our Pythia results: scaling behaviors of attack and defense are similar between the new model and Pythia, though robustness from model size alone is less clear than with Pythia.*

**Improve the motivation and framing of our experiments in the paper**

We have restructured the framing of the paper to center around our guiding question: in the context of language model adversarial robustness, what will the world look like when both attacker and defender have access to more compute? We make clear that we are not trying to improve defenses or attacks per se, but rather study the scaling behavior of generalizable attacks and defenses across a range of model sizes and scales of compute.

**Replicate our classification experiments using the Qwen2.5 model family on the Spam and Harmless tasks to ensure that our results generalize to a more modern model family**

We have done this, and show results in Figure 2, Figure 3, Figure 4, and Figure 5.

We observe the following:

* Figure 2: as with Pythia, with Qwen we do not observe a consistently improving robustness as model size increases. With Pythia, we were able to study models over 3 orders of magnitude, and observed some positive correlation between model size and robustness, though it was a weak effect and did not apply to all tasks. No such positive trend is visible with Qwen although it is possible we would observe a similar trend if Qwen came in smaller model sizes.

* Figure 3: note that the figure in the current version of the paper had a bug where we swapped the 3B and 7B model sizes when plotting Qwen; the [plot linked here](https://pasteboard.co/b4BCI506AKRe.png) fixes this issue. As with Pythia, we find that attack success rate on Qwen scales smoothly as a function of attack compute. We also notice a slight advantage of larger models over smaller models (ignoring the 0.5B model which we suspect received significantly more pretraining), though the effect is less pronounced than with Pythia (again, we suspect this is largely due to Qwen models spanning a limited range of sizes).

* Figure 4: as with Pythia, after adversarial training, attack success rate on Qwen scales smoothly and with similar slope across model sizes. Again overlooking the 0.5B model, we observe that across attack strengths the two larger models (3B and 7B) are harder to attack than the smaller model (1.5B). As with Figure 3, due to swapped model sizes, the version in the paper is slightly incorrect; see [here for the corrected version](https://pasteboard.co/8PNisTzM6IUi.png).

* Figure 5: As with Pythia, adversarial training smoothly improves defense for all Qwen model sizes, with models of different sizes all improving at approximately the same rate.

(continued)

---

> ### Author Response · Authors · 2024-11-30
> **(2/2)**
>
> (continuing)
>
> **(stretch) Benchmark the Qwen2.5 model family on StrongReject, to confirm that our results hold in the generative setting**
>
> We have done this with the Qwen2.5 Instruct models, and show results in Figure 2. We observe that robustness monotonically improves with model size. We believe that this trend is more clear than in the Qwen classification setting because of the instruction finetuning process, which likely includes something like safety training. While we do not have the details of the instruction finetuning, we suspect that the Qwen2.5 team performed instruction finetuning with comparable amounts of data across model sizes (and not comparable amount of compute across model sizes), which would lead to larger models becoming robust faster due to greater sample efficiency, as is the case when we plot adversarial robustness as a function of adversarial training round (eg, compare green line in the right plot of Figure 2 (also, [here](https://pasteboard.co/tmLS89Ty4xdD.png)) with one of the middle lines in the top left plot of Figure 23 (also, [here](https://pasteboard.co/Ugtyg0NUzbPB.png))—bigger models benefit more from sample-equivalent amounts of safety training).
>
> **(stretch) Evaluate our models with an additional adversarial attack (likely BEAST), to confirm that our results hold for a state-of-the-art black-box attack**
>
> We have done this, and show results in Figure 3. In our experiments, BEAST performs similarly (perhaps very slightly weaker, as one would expect) to GCG (note the different x and y limits on the plots in Figure 3).
>
> **We aim to complete these four tasks during the rebuttal period, though are less confident in our ability to do the last two in this timeframe, and might only have results after the end of the period. We will keep the reviewers updated as to our progress.**
>
> We have completed all four tasks (including two stretch goals) and look forward to continuing the conversation with reviewers. Thank you.
>
> ### Notes
> * Note that in estimating FLOP count, we missed three zeros for Qwen, so all x-axes in the current version of the paper that are measured in terms of “proportion of pretraining” for Qwen should be three orders of magnitude smaller (eg, 1e-2 should be 1e-5, etc).
> * As noted in the previous comment, we accidentally swapped the 3B and 7B model sizes when calculating proportion of pretraining compute, which changes the relative positions of the curves (but not the slopes).
> * See links in the first comment for examples of corrected plots.

---

### Author Response · Authors · 2024-12-04
**Thank you to the reviewers**

Thank you to the reviewers for your helpful feedback and engagement during the rebuttal period, we believe it has meaningfully improved the presentation, results, and impact of our paper.

---

### Meta-Review · Area_Chair_Q4Z7 · 2024-12-20

**Metareview:**

This paper investigates the relationship between model size scaling and robustness against adversarial attacks, with a primary focus on binary classification tasks. Through experiments conducted on the Pythia language model, the authors draw several conclusions regarding the robustness of large language models (LLMs) as their sizes change.

Reviewers acknowledge the empirical observations and analyses presented in the paper (all reviewers), the quality of the writing (Reviewers qvqy, zMEX), and the study of LLM scaling with adversarial training (Reviewers Vfij, EWYP, qvqy). However, significant concerns were also raised by most reviewers. These include an insufficient number of models evaluated (Reviewers qvqy, 4pZx, Vfij), narrowly designed experiments limited to simple tasks and a constrained range of hyperparameters (Reviewers qvqy, zMEX, EWYP, 4pZx, Vfij), and findings that are either unoriginal or overly dependent on specific tasks (Reviewers qvqy, EWYP).

During the rebuttal, the authors included new results to address the concerns raised, incorporating evaluations with additional LLMs, one new attack method, and one generative task. Despite the authors' significant efforts, the reviewers stated that many concerns remain inadequately addressed. The Area Chair also carefully reviewed the paper and all discussions. Taking into account the reviewers' opinions, this paper requires further improvement and cannot be accepted at this time.

**Additional Comments On Reviewer Discussion:**

The main concerns of Reviewer Vfij include the limited number of evaluated LLMs, simplistic evaluation tasks, and incomplete evaluation of adversarial training. In response, the authors provided new results on additional LLMs and introduced a text generation task. However, since Reviewer Vfij did not engage in the discussion after the authors’ rebuttal, their opinion is given less weight in the final decision.

Reviewer 4pZx expressed significant concerns regarding the experimental setup. Despite the authors’ new results, the reviewer did not respond to indicate whether their concerns were adequately addressed. The authors’ response addressed some of Reviewer EWYP’s concerns, including dataset selection and the inclusion of additional attack methods. Reviewer zMEX raised further concerns about the experimental design and the overall contribution of the paper. Finally, Reviewer qvqy maintained their concerns after the rebuttal, highlighting issues such as the experimental design, unoriginal claims, limited impact, and odd baselines.

Given the remaining unaddressed concerns raised by the reviewers, this paper requires further improvement and cannot be accepted at this time.

---

### Decision · Program_Chairs · 2025-01-22

Reject